# Impact of Canine Amniotic Mesenchymal Stem Cell Conditioned Media on the Wound Healing Process: In Vitro and In Vivo Study

**DOI:** 10.3390/ijms24098214

**Published:** 2023-05-04

**Authors:** Filip Humenik, Marcela Maloveská, Nikola Hudáková, Patrícia Petroušková, Zuzana Šufliarska, Ľubica Horňáková, Alexandra Valenčáková, Martin Kožár, Barbora Šišková, Dagmar Mudroňová, Martin Bartkovský, Daša Čížková

**Affiliations:** 1Centre of Experimental and Clinical Regenerative Medicine, The University of Veterinary Medicine and Pharmacy in Kosice, 040 01 Kosice, Slovakia; 2Small Animal Clinic, The University of Veterinary Medicine and Pharmacy in Kosice, 040 01 Kosice, Slovakia; 3Institute of Microbiology and Immunology, The University of Veterinary Medicine and Pharmacy in Kosice, 040 01 Kosice, Slovakia; 4Department of Food Hygiene, Technology and Safety, The University of Veterinary Medicine and Pharmacy in Kosice, 040 01 Kosice, Slovakia

**Keywords:** mesenchymal stem cell, amnion, conditioned media, wound healing, fibroblast

## Abstract

The aim of this study was to provide a beneficial treatment effect of mesenchymal stem cell products derived from the canine amniotic membrane (AM-MSC) on the complicated wound healing process in dogs. AM-MSCs were characterized in terms of morphology, phenotypic profile, and multilineage differentiation potential. The in vitro study of the effect of canine amniotic mesenchymal stem cell conditioned media (AMMSC-CM) on a primary skin fibroblast cell culture scratch assay showed a decrease in the measured scratch area of about 66.39% against the negative control (Dulbecco’s Modified Eagle’s Medium—32.55%) and the positive control (Dulbecco’s Modified Eagle’s Medium supplemented with FGF2, N2, B27, and EGF—82.077%) after 72 h treatment. In the experimental study, seven dogs with complicated nonhealing wounds were treated with a combination of antibiotics, NSAIDs, and local AMMSC-CM application. After 15 days of therapy, we observed a 98.47% reduction in the wound surface area as opposed to 57.135% in the control group treated by conventional therapy based on debridement of necrotic tissue, antibiotic therapy, pain management, and change of wound dressing.

## 1. Introduction

The occurrence of large-scale and chronic skin defects in companion animals is a frequent problem to solve. From an etiological point of view, we divide the provoking causes of chronic wounds into traumatic or iatrogenic. There are other important factors to consider when evaluating or treating a patient. After a traumatic insult, a wound may be just one of many injuries; therefore, it is important to perform a thorough general examination of the patient [1]. Healing of extensive skin defects caused by a traumatic insult is often challenging due to the need for complex management therapy consisting of targeted skin cleansing and removal of necrotic tissues, local or systemic control of infections, protection of surrounding tissue structures, and promotion of skin tissue regeneration [2]. In most cases, traumatic wounds are highly contaminated, which requires a special approach.

The goal of the chosen therapy is to close the skin defect as gently and as quickly as possible. To achieve this, a combination of surgical tissue reconstruction (i.e., skin graft) and medical treatment is often necessary to achieve complete anatomical and functional recovery of the damaged area [2,3]. In chronic wound therapy, conservative management is usually chosen, ensuring the support of the individual healing phases. Wound healing is a complex biological process, consisting of a series of consecutive events; therefore, it is recommended to focus on the intermingling of individual phases and the formation of new cells closing the skin defect [4].

The cellular and molecular mechanisms supporting tissue repair are still insufficiently understood [5]. One of the possibilities for influencing the formation and regeneration of cells is the application of adult mesenchymal stem cells (MSCs) locally to the wound surface. MSCs are multipotent cells capable of differentiating into any type of body cell under suitable conditions [6]. The most common sources of MSC acquisition are adipose tissue (AT) and bone marrow (BM); however, these cells can also be obtained from tissues and organs such as skin, bone, dental pulp, liver, ovarian epithelium, and others [4,6,7,8].

MSCs can differentiate into cells of mesodermal origins, such as chondrocytes, osteocytes, and adipocytes, into cells of the ectoderm, such as cells of nervous tissue, or into cells of endodermal origin [9]. The paracrine effect of stem cells also plays a significant role, since the production of significant growth and trophic factors or bioactive molecules affects angiogenesis, neurotrophicity, and immunoregulation in the target tissue [10]. It is also possible to explain the effect of a conditioned medium on angiogenesis. The products contained in the conditioned medium (CM) of MSC-like growth factors, cytokines, apoptotic bodies, and extracellular vesicles (EVs) positively affect the formation of new blood vessels and the growth of existing blood vessels in the system we are monitoring. The latest studies also describe the positive angiogenic effect of MSC products [11,12,13,14,15]. The mechanism of influencing angiogenesis is conditioned by the production of enzymes (PKA), the activation of some metabolic pathways (MAPK, PI3K/Akt), and the growth factors IGF-1, VEGF, von Willebrand factor, PDGF, PLGF, and TGF-ß [11]. During these processes, growth factors, cytokines, matrix metalloproteases, and angiogenic factors play an important role in processes such as inflammation, granulation tissue formation, re-epithelialization, matrix formation, and remodeling [16,17].

The aim of this study was to isolate and characterize MSCs from canine amniotic tissue according to the essential criteria of the International Society for Stem Cells Research (ISSCR). Subsequently, we prepared a conditioned medium from a cultivated population of AM-MSCs and compared its impact on the wound healing process in vitro (scratch assay performed on canine primary fibroblast culture) and in vivo (dogs with wounds of different etiology).

## 2. Results

### 2.1. Isolation of Canine MSCs from Amnion

Using the optimized protocol, we were able to isolate and cultivate a homogeneous population of canine MSCs from amnion (AM-MSCs). The yield of isolated cells from harvested tissue varied among 5–7 × 10^6^ cells/mL. Amniotic MSCs showed a fibroblast-like shape, which is typical for MSCs (Figure 1).

### 2.2. CD Characterization of Canine Amniotic MSCs

Results of CD analyses (Figure 2) showed that cell passaging is a suitable tool for obtaining high homogeneity and uniformity of the population during the cultivation of MSCs, even in low passage (all results are from passage 2). AM-MSCs showed high expression of CD29 (97.5 ± 1.0%), CD44 (85.1 ± 0.7%), and CD90 (92.8 ± 0.9%) and low expression of CD34 (0.7 ± 0.5%) and CD45 (1.2 ± 0.3%). We observed a high percentage of autofluorescent cells in AM-MSC–PE (1.9%) and APC (4.0%). The gating strategy was forward/sideward scatter and sideward scatter/sideward scatter pulse height to eliminate debris and doublets (Figure 2B).

### 2.3. Multilineage Potential

Using a commercial multilineage differentiation kit and the recommended culture protocol and staining methods, we confirmed the ability of MSCs isolated from canine amnion to differentiate into osteocytes and chondrocytes and only a weak ability to differentiate into adipocytes (Figure 3).

### 2.4. Isolation and Characterization of Canine Dermal Fibroblast Primary Culture

Using a standard protocol, we were able to isolate and cultivate a homogeneous population of canine dermal fibroblasts. The yield of isolated cells varied between 1.2 and 2.3 × 10^6^ cells/mL. Cells from the cultivated population showed a spindle shape and 80 µm in size (Figure 4).

Collagen I, Collagen III, and Vimentin are widely accepted as fibroblast markers [15,16,18]. The cultivated population was positive for each of the mentioned markers (Figure 5).

### 2.5. Scratch Assay

Bradford measurement, as a quantitative method for analyses of conditioned media from canines (AMMSC-CM), showed concentrations of proteins 1.75 mg/mL. For the scratch assay experiment, AMMSC-CM were diluted to a final concentration of 1.0 mg/mL.

The results from the scratch assay were obtained using Fiji (ImageJ) software, and its plugin Wound Healing Size Stack Tool showed that AMMSC-CM had a positive effect on the fibroblast experimental scratch healing model, since we observed a significant (*p* < 0.001) decrease in the scratch area from 23.52% to 7.903% after 72 h (Figure 6A,A1 and Figure 7). This observation is in contrast to the negative control (DMEM medium), in which the scratch area decreased from 23.587% to 15.908% (Figure 6B,B1 and Figure 7). However, the best results were observed in the positive control group (DMEM supplemented with B27, N2, recombinant human bFGF, and human EGF), in which the scratch area decreased significantly (*p* < 0.001) from 23.665% to 4.241% (Figure 6C,C1 and Figure 7).

### 2.6. Pilot Clinical Study: The Impact of Canine AMMSC-CM on Wound Healing

The percentage of surface area reduction (PAR) was significantly higher (*p* < 0.05) in the experimental group treated by AMMSC-CM with antibiotics and NSAIDs (ΔPAR = 86.34% observed on the 10th day and 98.47% on the 15th day) than in the control group of dogs not treated with AMMSC-CM (ΔPAR = 51.74% observed on the 10th day and 57.135% on the 15th day) (Figure 8). Representative photo-documentation from the wound healing process is presented in Figure 9.

## 3. Discussion

The main goal of this study was to observe the effect of CM from canine AM-MSCs on healing wounds in both in vitro and in vivo conditions.

For the isolation of MSCs from canine amnion, we used a combined method of mechanical disruption and enzymatic digestion. As a previous study has described, prolonged digestion can damage the cells [19]. Therefore, we optimized the time of the enzymatic process to 35–40 min, depending on the amount and size of the digested fraction at a temperature of 37 °C. The yield of AM-MSCs isolated by using this method was 1–1.6 × 10^6^ cells/gram, which correlated with other studies that have described a yield of isolation of 1–2 × 10^6^ per gram of human amniotic tissue [20,21]. However, it is still less than the isolation yield of AT-MSCs (2.5 × 10^6^ cells/g), but higher than the isolation yield of BM-MSCs (1 × 10^3^ cells/mL) confirmed by other studies [22,23]. MSCs isolated from canine amniotic tissue show a fibroblastoid-like shape and a size of 120–190 μm. We observed large differences in multilineage capability. Canine MSCs isolated from amniotic tissue show the high capability of osteogenic and chondrogenic potential. However, we confirmed only poor and nontypical adipogenic potential, even by using different commercial kits and times of cultivation and after repetition of the experiment. The same facts were described by other studies [8,24,25]. These circumstances could be explained by the addition of osteogenesis-mediating and adipogenesis-suppressing bioactive molecules, such as Runx2, Wnt10b ALP, OSX, and RhoA; adipogenesis-enhancing bioactive molecules suppressing osteogenesis, including PPARγ, P2 × 6, LIF, sFRP-1, and BMPs, also play an important role [26,27]. Thus, differentiation to mesoderm cell lines (osteocytes, chondrocytes, fibrocytes, myoblasts) is the key property required for hard and soft tissue regeneration in veterinary medicine [28,29].

The expression of CD surface markers in MSC populations was tested using flow cytometry. During the analysis, we found consistent expression of CD29, CD44, and CD90 (90% ± 2.5%); almost no cells expressed CD34 and CD45. These results correlate to the previous multilineage potential assay, in which we described poor adipogenic potential because the high percentage of CD29 and CD90 positive cells in the population reduces adipogenic activity [30]. Next, high expression of CD29 and CD44 is superior to strong chondrogenic potential [31]. For our experiment, the expression of CD90/Thy-1 plays an important role, because it is implicated in angiogenesis and blood perfusion during wound closure [32]. Concluding from the above-mentioned facts, amnion tissue represents a relatively rich source of stem cells. The process of harvesting and isolation of AM-MSCs did not cause any added stress or suffering for the animal, and since they have no further use in common veterinary practice, they would end up as biological waste. Therefore, this tissue source certainly deserves attention in the field of regenerative medicine.

MSCs play an important role in the wound- healing process. They participate in the homeostasis phase by promotion of coagulation due to the high content of tissue factor and phosphatidylserine [33]. MSCs modify the inflammatory phase by the production of bioactive molecules that polarize M1 proinflammatory macrophages to M2 anti-inflammatory type or suppress the effect of NK cells, which is a key role in wound healing and inflammation control [34]. According to our results, conditioned media of MSCs enhance the proliferative activity and migration of fibroblasts in vitro. Very similar results were published by Pommato et al., where EVs of AT-MSCs and BM-MSCs were used to study beneficial effects on the proliferation capacity and migration of fibroblasts, keratinocytes, and endothelial cells [14]. The mechanism of this activity could be explained by the release of growth factors such as EGF, bFGF, and TGF that increase the proliferation and migration of fibroblasts [35,36]. Very similar results were published by Liu et al. from their study focused on the migration and activation of human skin fibroblasts affected by MSC conditioned medium [37]. According to the above-mentioned facts, we confirmed that MSCs improve the proliferative phase of wound healing as well. The maturation phase is known as the last step of the wound repair process. MSCs attend in this phase by the production of numerous soluble factors and cytokines that suppress myofibroblast differentiation, enhance epithelial-mesenchymal transition, or show anti-fibrotic and proangiogenic potential (hepatocyte growth factor, prostaglandin E2, IL-10 adrenomedulin, vein endothelial growth factor, epidermal growth factor, or CXCL2) [38,39,40]. In our study, we used canine AMMSC-CM to affect the healing process of wounds in dogs. Our results show that MSC conditioned media reduced the wound area by about 98.47% on the 15th day when compared to the control group, where, on the same day, the wound area was reduced by about 57.135%. Many clinical studies on wounds have shown that the use of MSC therapy alone or in combination with other compounds such as platelet-rich plasma (PRP), hyaluronic acid, and others can stimulate the healing of skin defects [41,42,43,44]. Large skin lesions, where possible complications are poor blood circulation, tissue necrosis, excessive scarring, inflammation, and bacterial contamination, would benefit significantly from the application of regenerative therapy based on the synergistic action of PRP and MSCs [40]. The therapeutic effect against bacterial infections was also confirmed when MSCs were ingested. The antimicrobial properties of MSCs are probably due to their ability to reduce the inflammatory response and increase the phagocytosis of macrophages and monocytes. In addition, MSCs secrete GM-CSF, a cytokine that exhibits significant antimicrobial activity [45]. Similarly, the antimicrobial effect of mesenchymal stem cell conditioned media was published in our previous study [46].

A pilot study by Enciso et al. demonstrated higher regenerative capacity with earlier and faster closure of wounds treated using MSCs compared to other forms of treatment. In this study, the authors extended previous work to 24 dogs and evaluated the clinical relevance and safety of the application of adipose-derived allogeneic MSCs for acute and chronic skin wound healing [47]. In contrast to the aforementioned study, in our experimental treatment, we used acellular therapy with the application of stem cell products in the form of the conditioned medium. Similarly, Pomatto et al. show the positive effect of EVs isolated from ATMSC-CM and BMMSC-CM on healing of diabetic ulcers in mice [14]. The newest clinical study must be highlighted, with excellent results published by Gibello et al. that described the mechanism, safety, and validity of using EVs for the healing of chronic venous ulcers unresponsive to conventional treatments in humans [48]. These applications reduce the risk of an organism’s reaction after the application of stem cells and their subsequent separation.

## 4. Materials and Methods

First, we needed to obtain informed consent from the dogs’ owners, which is an essential criterion for approval of the study by the Ethical Committee of UVMP (EC UVMP) in Košice. The study was approved by EC UVMP on 2 September 2021 (EKVP/2021-01).

### 4.1. Isolation of MSC from Amniotic Tissue

Amniotic tissues were obtained during caesarean section (newborn puppies *n* = 11, 64th day of pregnancy) under strictly sterile conditions. The donor was Moscow Watchdog, the weight of puppies approximately 550 g, and the sex ratio 6:5. The amnions (amount of tissue 5 g) were washed with PBS (Biowest, Nuaillé, France) containing 2% antibiotic-antimycotic (ATB + ATM; penicillin-streptomycin-amphotericin B). The tissue was mechanically dissociated and enzymatically digested using 0.05% collagenase type IV at 37 °C for 30 min. At the end of the incubation period, the digested tissue was filtered (through a 100 μm cell strainer) to remove tissue fragments, and the obtained fraction was centrifuged at 400× *g* for 10 min. The obtained pellet was resuspended in Dulbecco’s Modified Eagle Medium/Nutrient Mixture F12 (DMEM-F12) culture medium + 10% Fetal Bovine Serum (FBS) + 2% ATB-ATM (Biowest). Cells were plated in a T25 culture flask at a concentration of 10^6^ cells/mL and incubated in culture medium (DMEM-F12) + 10% FBS + 2% ATB + ATM at 37 °C and 5% CO_2_. Nonadherent cells were removed, and the medium was subsequently changed twice a week.

### 4.2. Cell Passaging Procedures

When the cultivated cell population reached a confluence of approximately 75–80%, we proceeded to cell passaging. To separate the cells from the surface of the culture flask, an enzymatic trypsinization method using Trypsin EDTA 1× (Biowest), which acted on the cells depending on the level of confluence, was used for 5–7 min at 37 °C. To inactivate the trypsin, FBS in a 1:1 ratio was used, and the whole suspension was subsequently centrifuged at 400× *g* for 10 min. The supernatant was removed, and the cell population was plated on T75 culture flasks at a concentration of 1.5 × 10^6^ cells/flask.

### 4.3. CD Characterization of Amniotic MSCs

For the flow cytometry procedure, amniotic MSCs from passage 2 (P2) were used. Samples were analyzed for CD29 and CD44, CD90-positive and CD45 and CD34-negative cells. Each sample was diluted to a final concentration of 2 × 10^5^ cells and centrifuged at 400× *g* for 5 min. Subsequently, the supernatant was removed and the cell pellet was resuspended in 100 μL of PBS containing 3–5 μL of CD90 (YKIX337.217, monoclonal antibody, allophycocyanin (APC)), CD29 (MEM-101, monoclonal antibody, phycoerythrin (PE)), CD44 (MEM-263, monoclonal antibody, APC), CD34 (1H6, monoclonal antibody, PE), and CD45 (YKIX716.13, monoclonal antibody, PE) (all Thermo Fisher, Waltham, MA, USA) and incubated for 60 min at 4 °C in the dark. At the end of the incubation period, the samples were centrifuged again at 400× *g* for 5 min, the supernatant was removed, and the sample was washed in 200–500 µL of washing solution (1% BOFES in PBS + 0.1% Sodium Azid). Cytometric analysis was performed on a BD FACSCanto^®^ flow cytometer (Becton Dickinson Biosciences, East Rutherford, NJ, USA) equipped with a blue (488 nm) and red (633 nm) laser and 6 fluorescence detectors. The percentage of cells expressing individual CD markers was determined by a histogram for the respective fluorescence. The data obtained via measurement were analyzed in BD FACS Diva^TM^ analysis software. As a negative control, the same type of non-marked MSCs for the control of autofluorescence was used. The gating strategy for flow cytometry was performed by forward/sideward scatter and sideward scatter/sideward scatter pulse height to eliminate debris and doublets. The viability of observed cells varied between 85 and 96%.

### 4.4. Multilineage Potential

To confirm the multiline potential of the AM-MSCs, the MesenCult–ACF Chondrogenic Differentiation Kit MesenCult Osteogenic Stimulatory Kit and MesenCult Adipogenic Differentiation Kit (all STEMCELL Technologies, Vancouver, BC, Canada) were used according to the protocol instructions. The cells used for multiline differentiation were from P2. Cells were cultured in 24-well plates with an initial density of 5 × 10^4^ cells/well for osteocytes, 5 × 10^4^ cells/well for adipocytes, and 6 × 10^4^ cells/well for chondrocytes. Each micromass represented a single drop of 5 μL 6 × 10^4^ cells, which was placed in the center of the well and then incubated at 37 °C and 5% CO_2_ for 2 h for better adherence to the surface. Then, 500 μL of chondrogenic medium was added. After the recommended culture time (21 days), the cells were fixed using 4% paraformaldehyde (PFA), and the individual populations were stained with the Alizarin red staining method (Sigma, St Louis, MO, USA) for evidence of calcium deposits in the osteoblast population; Alcian blue (Sigma) for the detection of proteoglycans in the chondroblast population; and Oil red (Sigma) for the staining of fat vacuoles in the adipocyte population.

### 4.5. Preparation of Conditioned Media from AM-MSCs

The CM were prepared as described in our previous publication (Humenik et al. 2019 [12]. Shortly, AM-MSCs (P2) were cultured in DMEM (Biowest) without FBS (Biowest). After 24 h incubation in a humidified atmosphere with 5% CO_2_ at 37 °C, collected media samples were filtered through a 0.2 µm sterile syringe filter (Millipore, Burlington, MA, USA). To ensure that equal concentrations (2.0 mg/mL) of CM were used for the subsequent experiments, the protein concentration of the CM was quantified by Bradford protein assay using standard Bradford reagent (Sigma). As a control (nonconditioned medium), DMEM was regarded. Samples of AMMSC-CM were collected and stored at −80 °C until use.

### 4.6. Primary Culture of Canine Fibroblasts

Canine fibroblasts were isolated from skin samples (2 × 1 cm) of young, healthy dogs (n = 3) during routine gynecological surgery from the place of the incision. Donor 1: female, German shepherd, 35 kg, 3 years old. Donor 2: female, beagle, 17 kg, 4 years old. Donor 3: female, Hungarian pointer, 22 kg, 4 years old. After harvesting, samples were transported in DMEM medium with 2% ATB-ATM in sterile conditions. Skin samples were then washed in 70% ethanol solution and twice in PBS. In the next step, samples were cut into small pieces (0.5 × 0.5 cm). To divide the dermis and epidermis layer, an enzymatic method by Dispase II (Gibco, Billings, MT, USA) digestion in concentration 2.0 U/mL at 37 °C for 1 h was used. After dispase digestion, the dermis was mechanically separated from the epidermis. For the isolation of fibroblasts from the dermis, enzymatic digestion by Collagenase IV (Gibco) in concentration 0.05 mg/mL for 6 h at 37 °C was used. The cell suspension was centrifuged for 8 min at 400× *g*. Cells were plated in 6-well plates (3 × 10^5^ cells/well) and cultured in DMEM-F12 supplemented with 10% FBS, 2% ATB-ATM, bFGF (basic fibroblast growth factor, 20 ng/mL, Milipore), and EGF (epidermal growth factor, 20 ng/mL, AppliChem) at 37 °C in 5% CO_2_ incubator for 4 days in vitro (DIV4).

### 4.7. Immunocytochemistry Characterization of Canine Skin Fibroblasts

Canine skin fibroblasts (DIV4) were fixed with 4% PFA for 15 min and incubated with Anti-Vimentin primary antibody (Invitrogen; Abcam; MA5-11883; 1:200), Anti-Collagen I (rabbit polyclonal; Invitorgen; PA1-26204, 1:200), and Anti-Collagen III (mouse monoclonal; Invitrogen; CSI 007-01-02, 1:200). Cells were incubated with primary antibodies in PBS with Triton X-100 (0.1%) overnight. Secondary antibodies conjugated with FITC (green goat antimouse, Invitrogen, 62-6511, 1:500 and goat antirabbit, Invitrogen, F-2765, 1:500) were used. The nuclei were stained with 4, 6–diamidino-2-phenylindole (DAPI, Sigma, Cibolo, TX, USA). Incubation time for secondary antibody and DAPI was 1 h. The staining was detected by fluorescent microscopy (Zeiss, Germany), and pictures were taken with a fluorescent microscope camera (Zeiss Axiocam ERc 5s, Zeiss, Germany).

### 4.8. Scratch Assay

The scratch assay was performed on the characterized canine fibroblast population. When the primary fibroblast culture reached the confluence of 60–70% (DIV6), a 500 μm wide groove in the cell population was created using a 200 μL tip to simulate. The width of the groove was measured by Zeiss AxioVert software. In the experimental group, the culture medium with AMMSC-CM (n = 3) was replaced. DMEM medium without supplements was used as a negative control (n = 3). As a positive control (n = 3), DMEM supplemented with B27 (10 ng/mL, Gibco), N2 (10 ng/mL, Gibco), recombinant human bFGF (20 ng/mL), and human EGF (20 ng/mL) was used. Scratch areas were checked at the moment of mechanical injury simulation (0 h) and after every 24 h for 3 days. Evaluation of the experiment was performed by Fiji (ImageJ) and its plugin Wound Healing Size Stack Tool, which measures the area of the scratch. Data analysis was carried out via GraphPad Prism v8.3. (GraphPad Software, San Diego, CA, USA). The results of each variable were expressed as mean value ± SEM. One-way analysis of variance (ANOVA) was used to evaluate statistical significance at each time point interval (24 h, 48 h, and 72 h) in comparison to the time point of the scratch (0 h) for each group, with a significance level set at *p* < 0.001.

### 4.9. Pilot Clinical Study: The Impact of Canine AMMSC-CM on Wound Healing

In the experimental wound healing group, 7 dogs with complicated wounds of different etiology (bites, trauma, section, postoperative complications, etc.) from the university veterinary hospital of UVMP in Košice were treated with AMMSC-CM after the failure of conventional surgery consecution. AMMSC-CM was applied on the surface of the wound after its dressing change in the amount of 1 mL/cm^2^ and gently overlapped with a jelly patch (Gelita-Spon Standard, Gelita Medical GMBH). The secondary layer of the outer covering was synthetic cotton wool (Cellona, Lohman and Raucher int., Germany), and the tertiary layer and fixation of the bandage were ensured by applying a self-adhesive covering (Copoly, M+ H Vet). Re-bandages were performed at an interval of every 72 h, taking into account the condition and progress of wound closure. At the stage of 90% closure of the wound, yellow petroleum jelly containing retinol acetate and ergocalciferol (Infadolan ung. Der., Herbacos Recordati) was applied. As a support therapy, patients were given a combination of NSAIDs ((meloxicam (0.04 mg/kg, Meloxidolor, Sevaron + metamizole (40 mg/kg, Novalgin, Sanofi Aventis)) and convenient antibiotic therapy, according to the results of bacteriological cultivation (Noroclav, 8.75 mg/kg, Norbrook; Convenia, 8 mg/kg, Zoetis). In the control group (4 dogs), wounds of similar character and area (anatomical localization of wounds—neck, abdomen, limbs) were healed by using the same method; however, AMMSC-CM was excluded. Treated wounds were checked and photo-documented on day 1, 3, 5, 10, and 15. Wounds were measured by using the planimetric method (Area of wound = I × w × π/4 (I–length of wound, w–width of wound)) on days 1, 3, 5, 10, and 15 of the treatment. Measurements were always performed by one person in duplicate. Next, the Percentage of Area Reduction (PAR) according to pattern PAR = [(initial area − final area)/initial area × 100] was calculated. Data analysis was carried out via GraphPad Prism v8.3 (GraphPad Software, San Diego, CA, USA). The results of each variable were expressed as mean value ± SEM. One-way analysis of variance (ANOVA) and Tukey’s test for multiple comparisons of means were used to evaluate statistical significance between control and experimental groups on each day of evaluation, with a significance level set at *p* < 0.01.

## 5. Conclusions

The present study describes the impact of canine amniotic mesenchymal stem cell conditioned media on a canine dermal fibroblast primary culture tested with scratch tests and experimental wound healing assay on dogs with complicated wounds of different etiology. We observed the enhancement of fibroblast activation and proliferation treated by conditioned media in vitro. Moreover, we confirmed the positive effect of the conditioned medium to increase the efficiency of the biological processes taking place in the wound healing process in dogs. In the end, it is necessary to say that the therapeutic effect of the conditioned medium of stem cells depends on the content of EVs and free molecules, which are its essential part.

## Figures and Tables

**Figure 1 ijms-24-08214-f001:**
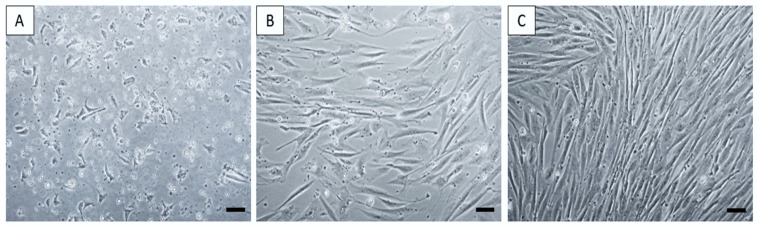
Morphology of canine MSCs from amniotic tissue. AM-MSC passage P0 at day in vitro (DIV) 4 (**A**), DIV7 (**B**), and DIV11 (**C**). Scale bars: 50 µm.

**Figure 2 ijms-24-08214-f002:**
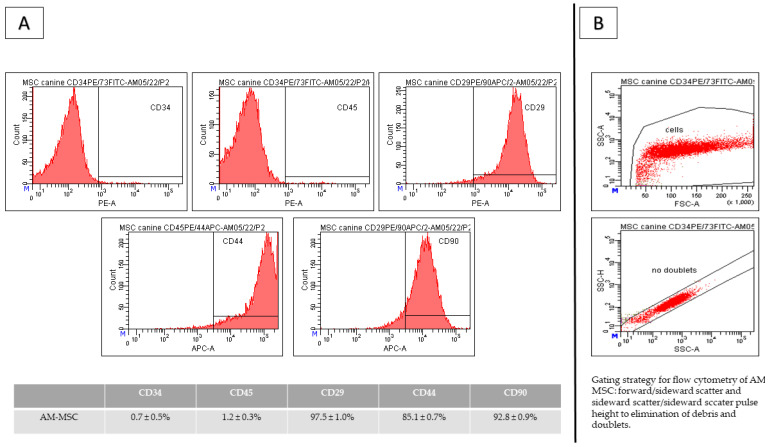
Results of CD analyses of canine amniotic MSCs (**A**) from passage 2 (P2) and gating strategy (**B**). AM-MSCs showed positivity for CD29 (97.5 ± 1.0%), CD44 (85.1 ± 0.7%), and CD90 (92.8 ± 0.9%) and low expression of CD34 (0.7 ± 0.5%) and CD45 (1.2 ± 0.3%). Gating strategy: forward/sideward scatter and sideward scatter/sideward scatter pulse height to eliminate debris and doublets.

**Figure 3 ijms-24-08214-f003:**
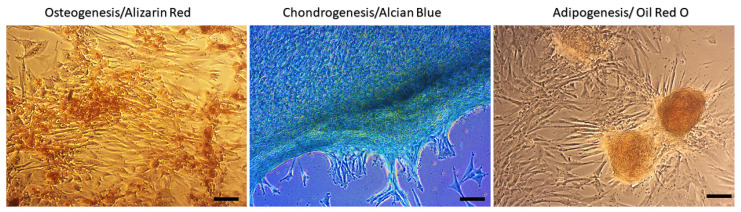
Multilineage potential of canine amniotic MSCs. Canine AM-MSCs showed high osteogenic (presence of calcium deposits detected by Alizarin red) and chondrogenic potential (presence of glycoproteoglycanes detected by Alcian blue staining); however, cells showed low adipogenic potential (triglycerides detected by Oil Red O staining). Scale bars: 50 µm.

**Figure 4 ijms-24-08214-f004:**
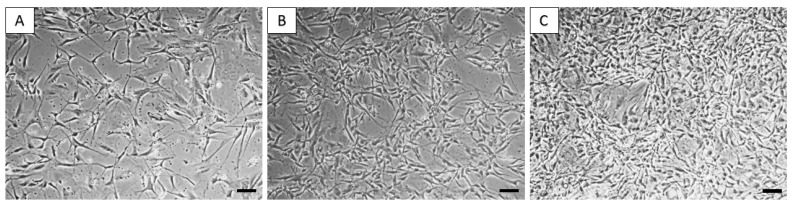
Morphology of canine dermal fibroblasts. Canine fibroblasts from passage P0 at DIV5 (**A**), DIV9 (**B**), and DIV15 (**C**). Scale bars: 50 µm.

**Figure 5 ijms-24-08214-f005:**
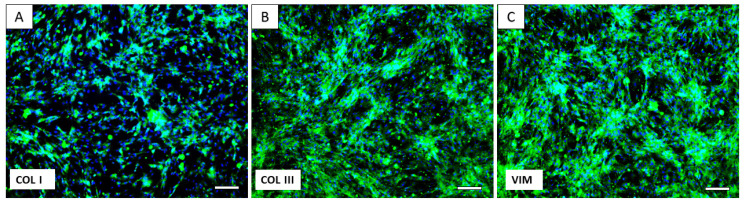
Immunocytochemistry characterization of canine dermal fibroblasts. Expression of Collagen 1 (**A**), Collagen III (**B**), and Vimentin (VIM) (**C**) detected by fluorescent microscopy. The nuclei were stained with DAPI. Scale bars: 100 µm.

**Figure 6 ijms-24-08214-f006:**
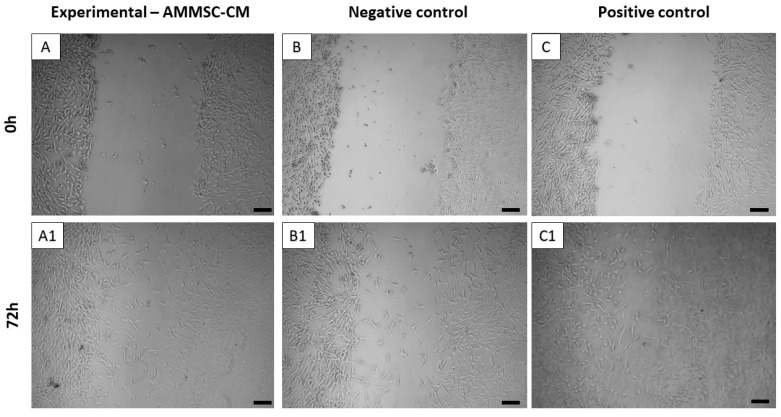
Scratch assay on fibroblast experimental scratch healing model. (**A**,**A1**) AMMSC-CM; (**B**,**B1**) negative control (DMEM, no supplements); (**C**,**C1**) positive control (DMEM supplemented with B27, N2, recombinant human bFGF, and human EGF). Brightfield images illustrate the decrease in the scratch area after 72 h of cultivation (**A1**–**C1**) in comparison to the initial time (0 h) of the mechanical injury simulation (**A**–**C**). AMMSC-CM demonstrated the regenerative potential on the canine fibroblast population. Scale bar: 50 µm.

**Figure 7 ijms-24-08214-f007:**
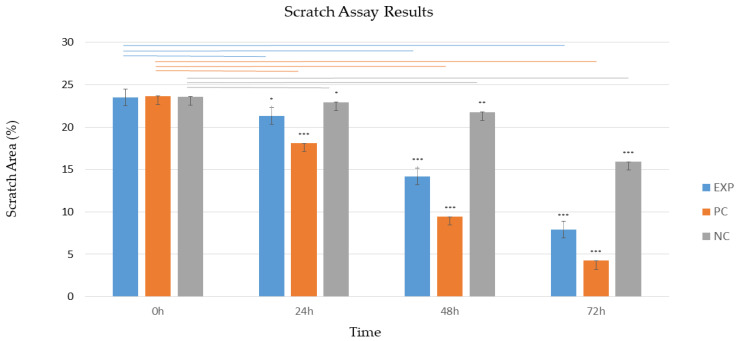
A statistical representation of the scratch assay on fibroblast experimental scratch healing model. The graph represents the scratch area measured after 24 h, 48 h, and 72 h of the mechanical injury simulation (0 h). Data represent mean value ± SEM. A statistically significant difference (*p*  <  0.001, one-way ANOVA, GraphPad Prism v8.3) of the regenerative potential was calculated at each time point interval (24 h, 48 h, and 72 h) in comparison to the time point of the scratch (0 h) for each group (AMMSC-CM—blue, positive control—orange, negative control—gray). * *p* < 0.05; ** *p* < 0.01; *** *p* < 0.001.

**Figure 8 ijms-24-08214-f008:**
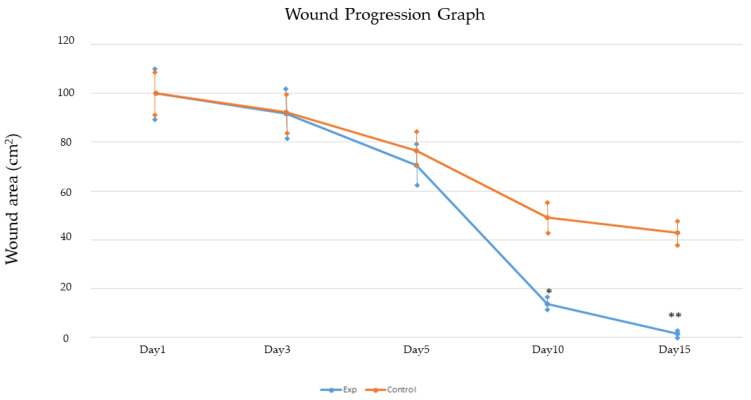
A statistical representation of the wound healing progression in the pilot clinical study. The graph represents the reduction in the wound area in the experimental group of dogs treated with AMMSC-CM with the addition of antibiotics and NSAIDs (Experimental—blue) and in the control group of dogs treated with conventional therapy (Control—orange). Data represent mean value ± SEM. A statistically significant difference (*p*  <  0.01, Tukey’s post hoc test, GraphPad Prism v8.3.) was calculated during the interval of 15 days of examination between the experimental and control group on each day of evaluation. * *p* < 0.05; ** *p* < 0.01.

**Figure 9 ijms-24-08214-f009:**
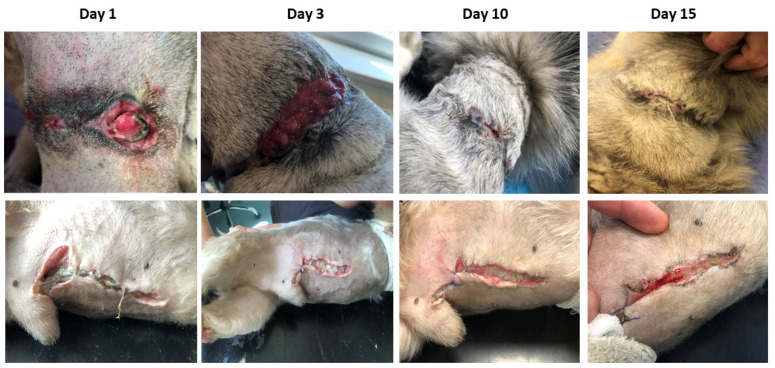
Wound healing progress in the pilot clinical study. The upper panels show the process of wound healing by using experimental treatment based on the combination of AMMSC-CM, antibiotics, and NSAIDs. Circular wound localized on the neck of dog. The lower panels show a wound of similar origin treated by the conventional method (antibiotics and NSAIDs). Wound localized on the ventral part of abdomen.

## Data Availability

The data presented in this study are available upon request from the corresponding author.

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
