# Peer review of "Impact of Canine Amniotic Mesenchymal Stem Cell Conditioned Media on the Wound Healing Process: In Vitro and In Vivo Study"

_ijms, 2023, doi:10.3390/ijms24098214_

Round 1

Reviewer 1 Report

 The study by Humenik et al. aimed to evaluate the therapeutic effect of CM derived from the canine amniotic membrane (AM-MSC) on the complicated wound healing process in dogs. They demonstrated both in vitro and in vivo that the CM display healing properties.

Concerns

1.       The authors did not discriminate between soluble factors and the extracellular vesicles (EVs) when they looked at the healing processes. The healing properties of EVs derived from different sources  including serum has been extensively reported. The authors have to perform at least in vitro studies to evaluate whether soluble factors of EVs are much more relevant for the healing process.

2.        Recently a pilot clinical study in humans using EVs from serum has provided efficacy in patients with  chronic venous ulcers. This study must be commented. Moreover, a study comparing the efficacy of different sources of EVs ( mesenchymal origin)  demonstrated differences in term of healing properties (Pomatto et al.). In the discussion section  preclinical and clinical results must be commented.

3.       Several references lack the author’s name. Please check.

Author Response

Revision 1

Reviewer 1

Dear Reviewer,

               Please find enclosed a revised manuscript with point-by-point responses to your comments. We are thankful for your valuable and constructive comments and suggestions. We appreciate the time and effort that you dedicated to providing valuable feedback on our manuscript. We have incorporated the changes to reflect most of your suggestions. Modifications made in the revised manuscript are marked up using the “Track Changes” on.

The authors have made all possible required improvements to the manuscript as suggested. However, some aspects to be revised remain in the text.

  1. The authors did not discriminate between soluble factors and the extracellular vesicles (EVs) when they looked at the healing processes. The healing properties of EVs derived from different sources  including serum has been extensively reported. The authors have to perform at least in vitro studies to evaluate whether soluble factors of EVs are much more relevant for the healing process.

Response: We fully agree with the statement that EVs play an important role in wound healing. However, the isolation and characterization of EVs requires high-end laboratory equipment, which our center does not have. That is why we decided to design an experimental study based on the observation of effect  of conditioned medium, that shoud contain EVs, on the wound healing process. In the future, we will definitely be looking for an institution that can help us isolate EVs, and it is possible that the design of the next study will be based on a comparison of the effect of EVs and conditioned media at least in vitro.

  1. Recently a pilot clinical study in humans using EVs from serum has provided efficacy in patients with  chronic venous ulcers. This study must be commented. Moreover, a study comparing the efficacy of different sources of EVs (mesenchymal origin)  demonstrated differences in term of healing properties (Pomatto et al.). In the discussion section  preclinical and clinical results must be commented.

.      Response: All two mentioned studies were discussed in Part „1. Introduction“, please see the lines L69-L73 and in Part „3. Discussion“, please the lines L224-L228 and L260-L265.

  1. Several references lack the author’s name. Please check.

       Response: All references were checked and modified. Please see the lines L435-L556.

Reviewer 2 Report

Manuscript ID: ijms-2315009

Comments to the Authors

In the manuscript entitled “Impact of Canine Amniotic MSCs Conditioned Media on the Wound Healing Process: In Vitro and In Vivo Study”, authors investigated the effects of AMMSC-CM on wound healing. It is interesting that CM enhances wound healing in vitro and in vivo study. However, there are few issues that I feel need to be addressed.

1)      In figure 3, it is hard to observe the results of multi-differentiation by staining. Please confirm the results. I recommend PCR or quantitative analysis can support the results.

2)      In figure 7, please indicate multiple comparison of the results.

3)      In figure 5, please show error bar of the results. How many did authors perform for the results?

Author Response

Dear Reviewer,

               Please find enclosed a revised manuscript with point-by-point responses to your comments. We are thankful for your valuable and constructive comments and suggestions. We appreciate the time and effort that you dedicated to providing valuable feedback on our manuscript. We have incorporated the changes to reflect most of your suggestions. Modifications made in the revised manuscript are marked up using the “Track Changes” on.

The authors have made all possible required improvements to the manuscript as suggested. However, some aspects to be revised remain in the text.

  • In figure 3, it is hard to observe the results of multi-differentiation by staining. Please confirm the results. I recommend PCR or quantitative analysis can support the results.

Response: We fully agree that it would be more appropriate to use the PCR method to confirm multilineage potential. However, the International Society for Stem Cell Research does not provide an exact methodology for determining multilineage potential. Moreover, in our conditions of public procurement, procurement and ordering of primers and reagents would take a lot of time (6-8 weeks) and the analysis itself (isolation, cultivation and characterization of MSC + PCR) would take at least 2-3 weeks. However, for the sake of argument, I have to mention that in all our published studies we used commercial kits for the confirmation of multilineage potential of MSC. However, I appreciate your comment and I will perform the next analysis using the PCR method.

  • In figure 7, please indicate multiple comparison of the results.

Response: Fig 7. was modified. Please check the fig. 8 in text.

3)      In figure 5, please show error bar of the results. How many did authors perform for the results?

Response: You, probably meant Fig. 7. Fig. 7 was modified and error bars were added. Same in part „Methodlogy – 4.9.“, sentence about   measurment was added. Please, see the lines L404-L405.

Round 2

Reviewer 1 Report

The authors have almost completely addressed my concerns. In the case they did not they gave me a rational explanation. However, they have at least to say that CM contains both EV and free molecules accounting for the therapeutic effects. 

Author Response

Dear Reviewer,

               First of all, we would like to thank you very much for your constructive criticism and comments on the presented study. Please find enclosed a revised manuscript with response to your comment. We appreciate the time and effort that you dedicated to providing valuable feedback on our manuscript. We have incorporated the changes to reflect most of your suggestion. Modifications made in the revised manuscript are marked up using the “Track Changes” on.

  1. The authors have almost completely addressed my concerns. In the case they did not they gave me a rational explanation. However, they have at least to say that CM contains both EV and free molecules accounting for the therapeutic effects. 

Response: We agree with the opinion that the conditioned medium contains EVs and free molecules, which are responsible for the therapeutic effectiveness of the studied conditioned medium, therefore we also stated this statement in section “5. Conclusions”. Please, see the lines L419-L421 of revised manuscript.

Reviewer 2 Report

This paper reports impact of Canine amniotic MSCs CM on the wound healing. The results showed that CM could be used as promising tools for wound healing. Although the results are of interest, the submitted manuscript should be more revised for publication. I am extremely sorry for the negative decision.

Author Response

First of all, we would like to thank you very much for your constructive criticism and comments on the presented study. Please find enclosed a revised manuscript with response to your comment from Revision report 1 and Revision report 2. We are deeply sorry that we made several mistakes in the pre-send of the answer, primarily related to the labeling of images. We appreciate the time and effort that you dedicated to providing valuable feedback on our manuscript. We have incorporated the changes to reflect most of your suggestion. Modifications made in the revised manuscript are marked up using the “Track Changes” on.

Revision report 1

  • In figure 3, it is hard to observe the results of multi-differentiation by staining. Please confirm the results. I recommend PCR or quantitative analysis can support the results.

Response: We fully agree that it would be more appropriate to use the PCR method to confirm multilineage potential. However, the International Society for Stem Cell Research does not provide an exact methodology for determining multilineage potential. Moreover, in our conditions of public procurement, procurement and ordering of primers and reagents would take a lot of time (6-8 weeks) and the analysis itself (isolation, cultivation and characterization of MSC + PCR) would take at least 2-3 weeks. However, for the sake of argument, I have to mention that in all our published studies we used commercial kits for the confirmation of multilineage potential of MSC. However, I appreciate your comment and I will perform the next analysis using the PCR method.

  • In figure 7, please indicate multiple comparison of the results.

Response: Fig 7. was modified. Please check the Fig. 7 in the text.

3)      In figure 5, please show error bar of the results. How many did authors perform for the results?

Response: You, probably meant Fig. 8. Fig. 8 was modified and error bars were added. Same in part „Methodlogy – 4.9.“, sentence about principle of wound area measurment was added. Please, see the lines L404-L405.

 Revision report 2

  • This paper reports impact of Canine amniotic MSCs CM on the wound healing. The results showed that CM could be used as promising tools for wound healing. Although the results are of interest, the submitted manuscript should be more revised for publication. I am extremely sorry for the negative decision.

 Response: According to Revision report 1, we made all possible required changes in the paper. We were not able to make a comment regarding the use of the PCR method to confirm multilineage potential for the objective reasons mentioned above. We are sorry that we did not meet all your requirements, but nevertheless we hope for a positive opinion on the publication of the submited paper.